# Human β-Defensin 3 Inhibition of *P. gingivalis* LPS-Induced IL-1β Production by BV-2 Microglia through Suppression of Cathepsins B and L

**DOI:** 10.3390/cells13030283

**Published:** 2024-02-04

**Authors:** Erika Inoue, Shiyo Minatozaki, Sachi Shimizu, Sayaka Miyamoto, Misato Jo, Junjun Ni, Hidetoshi Tozaki-Saitoh, Kosuke Oda, Saori Nonaka, Hiroshi Nakanishi

**Affiliations:** 1Faculty of Pharmacy, Yasuda Women’s University, Hiroshima 731-0153, Japan; 19141105@st.yasuda-u.ac.jp (E.I.); 19141237@st.yasuda-u.ac.jp (S.M.); 20141218@st.yasuda-u.ac.jp (S.S.); 20141243@st.yasuda-u.ac.jp (S.M.); 20141219@st.yasuda-u.ac.jp (M.J.); 2Key Laboratory of Molecular Medicine and Biotherapy, School of Life Science, Beijing Institute of Technology, Beijing 100081, China; nijunjun@bit.edu.cn; 3Department of Pharmaceutical Sciences, School of Pharmacy at Fukuoka, International University of Health and Welfare, Okawa 831-8501, Japan; saitoh@iuhw.ac.jp; 4Department of Pharmacology, Faculty of Pharmacy, Yasuda Women’s University, Yasuhigashi, Hiroshima 731-0153, Japan; oda-k@yasuda-u.ac.jp (K.O.); nonaka-s@yasuda-u.ac.jp (S.N.)

**Keywords:** BV-2 microglia, cathepsin B, CA-074Me, human β-defensin 3, interleukin-1β, lipopolysaccharide, nuclear factor-κB, outer membrane vesicles, *Porphyromonas gingivalis*

## Abstract

Cathepsin B (CatB) is thought to be essential for the induction of *Porphyromonas gingivalis* lipopolysaccharide (*Pg* LPS)-induced Alzheimer’s disease-like pathologies in mice, including interleukin-1β (IL-1β) production and cognitive decline. However, little is known about the role of CatB in *Pg* virulence factor-induced IL-1β production by microglia. We first subjected IL-1β-luciferase reporter BV-2 microglia to inhibitors of Toll-like receptors (TLRs), IκB kinase, and the NLRP3 inflammasome following stimulation with *Pg* LPS and outer membrane vesicles (OMVs). To clarify the involvement of CatB, we used several known CatB inhibitors, including CA-074Me, ZRLR, and human β-defensin 3 (hBD3). IL-1β production in BV-2 microglia induced by *Pg* LPS and OMVs was significantly inhibited by the TLR2 inhibitor C29 and the IκB kinase inhibitor wedelolactonne, but not by the NLRPs inhibitor MCC950. Both hBD3 and CA-074Me significantly inhibited *Pg* LPS-induced IL-1β production in BV-2 microglia. Although CA-074Me also suppressed OMV-induced IL-1β production, hBD3 did not inhibit it. Furthermore, both hBD3 and CA-074Me significantly blocked *Pg* LPS-induced nuclear NF-κB p65 translocation and IκBα degradation. In contrast, hBD3 and CA-074Me did not block OMV-induced nuclear NF-κB p65 translocation or IκBα degradation. Furthermore, neither ZRLR, a specific CatB inhibitor, nor shRNA-mediated knockdown of CatB expression had any effect on *Pg* virulence factor-induced IL-1β production. Interestingly, phagocytosis of OMVs by BV-2 microglia induced IL-1β production. Finally, the structural models generated by AlphaFold indicated that hBD3 can bind to the substrate-binding pocket of CatB, and possibly CatL as well. These results suggest that *Pg* LPS induces CatB/CatL-dependent synthesis and processing of pro-IL-1β without activation of the NLRP3 inflammasome. In contrast, OMVs promote the synthesis and processing of pro-IL-1β through CatB/CatL-independent phagocytic mechanisms. Thus, hBD3 can improve the IL-1β-associated vicious inflammatory cycle induced by microglia through inhibition of CatB/CatL.

## 1. Introduction

Microglia-mediated neuroinflammation is an important component of Alzheimer’s disease (AD) pathogenesis and has been implicated in neurodegeneration [1,2,3]. Interleukin-1β (IL-1β) is a potent proinflammatory cytokine involved in many important cellular functions. The release of IL-1β is a critical step in inflammation through the induction of other proinflammatory cytokines and chemokines [4,5]. IL-1β is chronically upregulated in AD and believed to play a role in the vicious inflammatory cycle that drives AD pathology [6]. A two-step process is generally necessary for IL-1β production. The first step is the synthesis of pro-IL-1β, and the second step is the processing of synthesized pro-IL-1β. The TLR-dependent signals first activate nuclear factor-κB (NF-κB) to lead pro-IL-1β synthesis. The Nod-like receptor (NLR) family pyrin domain-containing 3 (NLRP3) inflammasome then undergoes post-translational modifications that license its activation. The activation signals activate the NLRP3 inflammasome with subsequent activation of pro-caspase-1, which in turn catalyzes the cleavage of pro-IL-1β. 

Activation signals are provided by a variety of stimuli and multiple molecular or cellular events, including K^+^ efflux, mitochondrial dysfunction with reactive oxygen species (ROS) generation, and lysosomal damage with cathepsin B (CatB; EC 3.4.22.1) leakage. Several studies have suggested a CatB/NLRP3/caspase-1-dependent pathway for pro-IL-1β processing in BV-2 microglia [7] and THP-1 cells [8], and the roles of multiple cathepsins in NLRP3 activation have also been implicated in macrophages/microglia [9,10]. However, we previously reported a potential NLRP3-independent role of autolysosomal CatB in pro-caspase-1 activation and subsequent IL-1β secretion by microglia following stimulation with chromogranin A [11,12]. Therefore, there are at least three different pathways for pro-IL-1β processing, with a special focus on the role of CatB, depending on the stimulating reagents and cell types: (1) a CatB/NLRP3/caspase-1-dependent pathway; (2) a CatB/caspase-1-dependent, but NLRP3-independent pathway; and (3) a CatB and other cysteine cathepsins/NLRP3/caspase-1-dependent pathway. In addition to pro-IL-1β processing, there is some evidence demonstrating that CatB is also involved in pro-IL-1β synthesis through the degradation of IκBα, an endogenous inhibitor of NF-κB in macrophages/microglia at the late stage of inflammation [9,13,14,15].

The emerging role of microbes and innate immune pathways in AD pathology suggests that antimicrobial peptides may be effective as early therapeutic intervention in future clinical trials [16,17]. The salivary proteome contains a complex mixture of over 45 antimicrobial proteins and peptides, including human defensins, histatins, and cathelicidin hCAP18/LL-37 [18]. Human β-defensins (hBDs) are small, cationic antimicrobial peptides produced by the oral mucosa and salivary glands. We previously reported that hBD3 strongly suppresses the delayed type of inflammatory responses by microglia following treatment with a lipopolysaccharide (LPS) derived from Porphyromonas gingivalis (*Pg*), a major pathogen of chronic periodontitis. hBD3 suppresses *Pg* LPS-induced NF-κB activation through the inhibition of CatB and cathepsin L (CatL; EC 3.4.22.15) [19]. Furthermore, we first reported that chronic systemic exposure to *Pg* LPS induces AD-like pathologies, including microglia-mediated neuroinflammation and cognitive decline in middle-aged mice, but not in CatB-deficient mice [20]. 

However, whether or not hBD3 and CatB inhibitors can suppress *Pg* virulence factor-induced IL-1β production by microglia remains unclear. In addition to *Pg* LPS, chronic oral gavage with outer membrane vesicles (OMVs) secreted from *Pg* also induced AD-like pathologies in middle-aged mice [21]. In the present study, we have thus attempted to clarify the effects of hBD3 and CatB inhibitors on IL-1β production by microglia following stimulation with *Pg* LPS and OMVs.

## 2. Materials and Methods

### 2.1. Reagents

hBD3, CA-074Me, KYT-1, and KYT-36 were purchased from the Peptide Institute Inc. (Osaka, Japan), and Z-Arg-Leu-Arg-α-aza-glycil-Ile-Val-OMe (ZRLR) was synthesized by Peptide Institute Inc. Standard *Pg* LPS was purchased from InvivoGen (San Diego, CA, USA). TAK-242 and C29 were purchased from MedChem Express (Monmouth Junction, NJ, USA). MCC950 was purchased from Adipogen Life Sciences, Inc. (San Diego, CA, USA). Wedelolactone was purchased from Tokyo Chemical Industry Co., Ltd. (Tokyo, Japan).

### 2.2. Cell Culture 

The BV-2 cells, a murine microglial cell line [22], and a well-accepted alternative to primary microglia [23,24], were used in this study. BV-2 microglia were cultured in Dulbecco’s modified Eagle’s medium (ThermoFisher Scientific, Waltham, MA, USA) supplemented with 5% fetal bovine serum (FBS), penicillin, and streptomycin. To establish NanoLuc (Nluc) probe-expressing cells (Nluc reporter BV-2 microglia), we infected the cells with a lentiviral vector carrying the Nluc probe and selected EGFP-positive cells, as described previously [25]. The luciferase activity was only induced when proteolytic pro-IL-1β processing occurred because the Nluc luciferase was fused with the sequence of IL-1β (*Il1b* 17-216), which has been subjected to proteolytic processing of proteases including caspase-1 [26]. Furthermore, the *C*-terminus of *Il1b* 17-216 was fused with two protein destabilization sequences (hCL1 and hPEST) (Figure 1A). Thus, the Nluc luciferase protein was retained only when the proteolytic IL-1β processing was successful.

### 2.3. The Measurement of the Luciferase Activity (RLU) 

Nluc reporter BV-2 microglia were plated in 96-well culture plates at a density of 3 × 10^4^ cells per well. After overnight culture, drug treatments were performed, and luciferase activity following treatment with *Pg* LPS (10 µg/mL) or OMVs (150 µg/mL) for 1 h was measured using a luminometer (GloMax; Promega Corp., Madison, WS, USA) with a Nano-Glo^®^ luciferase assay system (N1110; Promega Corp.) according to the manufacturer’s protocol. The luciferase activity (RLU) in BV-2 microglia induced by *Pg* LPS or OMVs was then measured. Each treatment was repeated in triplicate on the same plate, and at least three independent experiments were performed. 

### 2.4. Bacterial Culture and Isolation of OMVs

There are three types of gingipains: two arginine-specific proteases (RgpA and RgpB) and a lysine-specific protease (Kgp). In this study, we used *Pg* ATCC33277 (wild-type; WT), Kgp-deficient mutant strain (KDP129), and RgpA, RgpB and Kgp-deficient mutant strain (KDP136) were maintained, as described previously [27]. For immunostaining experiments, OMVs were isolated as described previously [28]. In other experiments, OMVs were prepared by ultracentrifugation as described previously [29]. 

### 2.5. Fluorescence Imaging of Cellular Localization of OMVs

Cy5-labelled OMVs prepared from the WT and KDP136 strains (150 µg of total protein/mL) were incubated with Nluc reporter BV-2 microglia (2 × 10^4^ cells/well). Cy5-labelled OMVs from the WT and KDP136 strains were prepared as described previously [27]. BV-2 microglia were then incubated with Acti-stain 488 phalloidin (an F-actin detecting dye, Cytoskeleton, Inc., Denver, CO, USA) and Hoechst 33342 (ThermoFisher Scientific) to visualize the cells. Fluorescence images were obtained using a confocal laser-scanning microscope (CLSM; FV1000, Olympus, Tokyo, Japan).

### 2.6. Trasfection of CatB shRNA

Nluc reporter BV-2 microglia were transfected to CatB shRNA or control shRNA lentiviral particle (Santa Cruz Biotechnology, Inc., Dallas, TX, USA) according to the manufacture’s transfection protocol. In brief, the Nluc reporter BV-2 microglia were cultured with complete optimal medium in a 12-well plate (3 × 10^5^ cells/well) 24 h prior to lentiviral infection. Media were removed from the plate wells and replaced with 1 mL of polyberene/media mixture (5 µg/mL) per well. CatB or control shRNA lentiviral particles were then added and incubated overnight. The cells were cultured for an additional 48 h in complete medium without polybrene. Stable clones expressing CatB shRNA (Nluc reporter CatB-knockdown BV-2 microglia) or control shRNA were selected using puromycin dihydrochloride (6 µg/mL).

### 2.7. Fluorescence Imaging of Enzymatic Activities of CatB and CatL 

BV-2 microglia were stained with the cell-permeable fluorescently labeled CatB substrate z-Arg-Arg-cresyl violet or CatL substrate z-Phe-Arg-cresyl violet, as described previously [19]. Fluorescence images were obtained using CLSM (FV1000, Olympus). 

### 2.8. Nuclear NF-κB p65 Translocation

BV-2 microglia were treated with *Pg* LPS (10 μg/mL) or OMVs (150 µg/mL) for 1 h in the absence or presence of hBD3 (1 µM) or CA-074Me (30 μM), and were fixed with 4% paraformaldehyde. hBD3 and CA-074Me were pre-incubated with BV-2 microglia for 2 and 4 h, respectively. Nuclear NF-κB p65 was detected by immunostaining performed as previously described [19]. The line plot profile was analyzed using the Image J 1.53 software program. 

### 2.9. Immunoblotting Analyses

For detecting IκBα and β-actin, Nluc BV-2 microglia were used. WT and CatB-KD Nluc BV-2 microglia were used to detect CatB and GAPDH. Cells were seeded in a 6 cm petri dish at a density of 2.5–3.3 × 10^6^ cells/dish for 1 day. After treatment with *Pg* LPS or OMVs in the absence or presence of hBD3 (1 μM) or CA-074Me (30 μM), the cells were subjected to immunoblotting analyses, as described previously [19,27]. 

### 2.10. AlphaFold Predictions 

The AlphaFold models were predicted using the AlphaFold v2.0 algorithm on the Colab server (https://colab.research.google.com/github/sokrypton/ColabFold/blob/main/AlphaFold2.ipynb, accessed on 20 November 2023) [30]. Predictions were performed with default multiple sequence alignment generation using the MMSeqs2 server, with 48 recycles and templates (homologous structures). The best of the five predicted models (rank 1) computed by AlphaFold was considered in the present work.

### 2.11. Statistical Analyses

Data are presented as the mean ± standard error (SE). Statistical analyses of the results were performed using a one-way analysis of variance (ANOVA) with a post hoc Tukey’s test and Student’s *t*-test using the GraphPad Prism8 (GraphPad Software, Inc., La Jolla, CA, USA) software package. *p* < 0.05 was considered to indicate statistical significance.

## 3. Results

### 3.1. Effects of Inhibitors for TLR4, TLR2, IκB Kinase, and NLRP3 Inflammasome on the IL-1β Production by BV-2 Microglia Following Stimulation with Pg LPS and OMVs

To assess the involvement of TLRs in IL-1β production by BV-2 microglia following stimulation with *Pg* LPS and OMVs, the effects of TAK-242- and C29-specific inhibitors of TLR4 and TLR2, respectively, on IL-1β production were examined by measuring the luciferase activities of the Nluc reporter BV-2 microglia. *Pg* LPS-induced luciferase activity was almost completely suppressed by C29 (100 µM), but not by TAK-242 (1 µM) (Figure 1B). OMV-induced luciferase activity was significantly but not completely suppressed by C29 (100 µM) (Figure 1C). TAK-242 (1 µM) had no significant effect. Furthermore, wedelolactone (30 µM), an IκB kinase (IKK) inhibitor, significantly and completely inhibited the mean luciferase activity induced by *Pg* LPS and OMVs (Figure 1D). In contrast, MCC950 (10 µM), an NLRP3 inflammasome inhibitor, had no effect on the mean luciferase activity induced by *Pg* LPS or OMVs (Figure 1E). 

### 3.2. Possible Involvement of Phagocytosis of OMVs by BV-2 Microglia in IL-1β Production

We next examined the possible phagocytosis and cellular localization of Cy5-labeled OMVs by BV-2 microglia. F-actin localized around the cell periphery. Cy5-labeled OMVs were phagocytosed by BV-2 microglia and accumulated as coarse granular aggregates, suggesting endosomal/ lysosomal localization (Figure 2A,C). Vertical optical sections of BV-2 microglia clearly showed the intracellular localization of OMVs. 

### 3.3. Possible Involvement of Gingipains in IL-1β Production by BV-2 Microglia Following Treatment with OMVs

After secretion through the type IX system, Kgp and Rgp attach to anionic LPS located on the outer membrane surface of *Pg* [31]. Therefore, in addition to LPS, gingipains attached to the surface of OMVs may also be associated with IL-1β production. KYT-1 (10 µM, an Rgp inhibitor), KYT-36 (10 µM, a Kgp inhibitor), and the combination thereof had no effect on OMV-induced luciferase activity in BV-2 microglia (Figure 2B). Furthermore, OMVs prepared from KDP129, a Kgp-deficient mutant strain, induced luciferase activity in BV-2 microglia to a level similar to that of the WT. In contrast, OMVs prepared from KDP136, a Kgp- and Rgp-deficient mutant strain, did not induce luciferase activity in BV-2 microglia. Furthermore, it was also noted that BV-2 micoglia did not phagocytose Cy5-labeled OMVs prepared from KDP136 (Figure 2C).

### 3.4. Effects of Pharmacological and Genetic Inhibition of CatB on IL-1β Production by BV-2 Microglia Following Stimulation with Pg LPS and OMVs

Next, we examined the effects of CA-074Me, hBD3 and ZRLR on the IL-1β production by measuring the luciferase activity. Both CA-074Me (10 µM) and hBD3 (1 µM) significantly inhibited luciferase activity induced by *Pg* LPS in BV-2 microglia (Figure 3A). However, ZRLR (10 µM), a membrane-permeable specific inhibitor of CatB [32], did not inhibit luciferase activity. CA-074Me (10 µM) also significantly inhibited OMV-induced luciferase activity in BV-2 microglia, but neither hBD3 (1 µM) nor ZRLR (10 µM) inhibited this activity (Figure 3B). 

We examined the possible inhibitory effects of CA-074Me and ZRLR on the enzymatic activities of CatB and CatL in BV-2 microglia using cell-permeable, fluorescently labeled substrates, z-Arg-Arg-cresyl violet and z-Phe-Arg-cresyl violet, respectively. The fluorescent cresyl violet group was designed to be dequenched upon cleavage of dipeptides by CatB or CatL. CA-074Me (10 µM) markedly reduced the enzymatic activity of both CatB and CatL in BV-2 microglia. In contrast, ZRLR (10 µM) markedly reduced the fluorescent signal of CatB, but not that of CatL (Figure 4). 

We further examined the effect of the genetic inhibition of CatB on the IL-1β production by BV-2 microglia following stimulation with *Pg* LPS and OMVs. As shown in Figure 3C, mature CatB was detected in a single-chain form and in the heavy chain of the two-chain form. The mean intensity of CatB immunoreactivity was significantly lower in BV-2 microglia after stable infection with CatB shRNA lentiviral particles than non-infected Nluc reporter BV-2 microglia (Figure 3C). The mean luciferase activity (RLU) of the IL-1β probe in BV-2 microglia following stimulation with *Pg* LPS and OMVs was 33 × 10^3^ (*n* = 3) and 21 × 10^3^ (*n* = 3), respectively (Figure 3D). The mean RLU measured in Nluc reporter BV-2 microglia (WT) following treatment with *Pg* LPS and OMVs was 15 × 10^3^ (range: 7–22 × 10^3^, *n* = 15) and 18×10^3^ (range: 8–26 × 10^3^, *n* = 21), respectively. Therefore, CatB knockdown did not affect IL-1β production by BV-2 microglia following stimulation with *Pg* LPS or OMVs.

### 3.5. Effects of CA-074Me and hBD3 on Pg LPS-Induced Nuclear NF-κB p65 Translocation in BV-2 Microglia Following Stimulation with Pg LPS and OMVs

The promoter regions of both IL-1β genes have a putative NF-κB binding site. Therefore, abrogation of NF-κB activation is a plausible mechanism underlying the inhibitory the effects of CA-074Me and hBD3 on pro-Il-1β synthesis. Thus, we examined the effects of CA-074Me and hBD3 on nuclear NF-κB p65 translocation following treatment with *Pg* LPS and OMVs. Both hBD3 (1 µM) and CA-074Me (30 µM) inhibited *Pg* LPS-induced nuclear NF-κB p65 translocation in BV-2 microglia (Figure 5A,B). In contrast, neither hBD3 (1 µM) nor CA-074Me (30 µM) inhibited OMV-induced nuclear NF-κB p65 translocation (Figure 5A,B). 

### 3.6. Effects of hBD3 and CA-074Me on the Degradation of IκBα Following Treatment with Pg LPS and OMVs 

Inhibitory effects of hBD3 and CA-074Me on the NF-κB activation induced by *Pg* LPS and OMVs were further evaluated by immunoblot analyses of IκBα degradation. The mean level of IκBα was significantly decreased in BV-2 microglia at 10–30 min after treatment with either *Pg* LPS (10 µg/mL) or OMVs (150 µg/mL). Both hBD3 (1 µM) and CA-074Me (30 µM) significantly inhibited IκBα degradation induced by *Pg* LPS in BV-2 microglia (Figure 6A,B). In contrast, neither hBD3 (1 µM) nor CA-074Me (30 µM) affected the IκBα degradation induced by OMVs (Figure 6C,D). 

The effects of the inhibitors used in this study on IL-1β production, nuclear NF-κB p65 translocation, and IκBα degradation by BV-2 microglia are summarized in Table 1.

### 3.7. Prediction of hBD3 Binding to CatB and CatL

To examine the possibility that hBD3 inhibits the activities of CatB and CatL through direct binding, a high-quality model of hBD3-bound CatB was generated using AlphaFold (Figure 7A,B). The CatB model in the complex had a median predicted Local Distance Difference Test (pLDDT) score of 95.54 for 257 Cα atoms, except for 3 residues at *C*-terminus (Glu258-Ile260) (Figure 7A). The hBD3 model in the complex had a median pLDDT score of 87.48 for 45 Cα atoms, which included 2 residues with low confidence in the loop between the β1 and β2 strands, Arg36 (68.00) and Gly37 (66.69) (Figure 7A). In addition, the predicted alignment error (PAE) between CatB and hBD3 was extremely low (Figure 7B). Therefore, the complex model is reliable for analyzing the interactions between proteins. In the CatB model, an active-site cleft was formed on the molecular surface. At the bottom, the substrate-binding pocket, consisting of S1–S3, S1’, and S2’ sites, were lined across the catalytic Cys29 residue (Figure 7C).

In the complex model, the first half (Cys18-Cys23) of the loop between the α1 helix and β1 strand of hBD3 was inserted into the cleft of CatB toward the S2 site, while the other half (Cys23-Lys26) was inserted toward the S3 site (Figure 7C). Therefore, the CatB:hBD3 complex model suggests that hBD3 may inhibit enzymatic activity by blocking the substrate-binding pocket in the cleft of CatB. 

On the other hand, the interaction between hBD3 and CatL, the model of hBD3-bound CatL indicates that hBD3 binds to the active cleft formed on the molecular surface of CatL (Figure 8A,B). However, the confidence level was relatively low (Figure 8C).

## 4. Discussion

hBD3 significantly suppressed IL-1β production induced by *Pg* LPS but not by OMVs, while CA-074Me significantly inhibited the IL-1β production induced by both *Pg* LPS and OMVs. Furthermore, both hBD3 and CA-074Me significantly inhibited *Pg* LPS-induced nuclear NF-κB p65 translocation and IκBα degradation. In contrast, neither hBD3 nor CA-074Me blocked OMV-induced nuclear NF-κB p65 translocation and IκBα degradation. CA-074 could have non-cathepsin off-target effects [9,34], which may be responsible for the inhibitory effect of CA-074Me on OMV-induced IL-1β production. On the other hand, ZRLR, a specific CatB inhibitor, had no effect on *Pg* virulence factor-induced IL-1β production. This was consistent with the results obtained using shRNA-mediated knockdown of CatB expression. These results suggest that *Pg* LPS induces the synthesis and processing of pro-IL-1β through CatB/CatL-dependent mechanisms, without activation of the NLRP3 inflammasome. In contrast, OMVs may promote the synthesis and processing of pro-IL-1β through CatB/CatL-independent phagocytic mechanisms. 

The structural models generated by AlphaFold in this study indicated that hBD3 can bind more strongly to the substrate-binding pocket of CatB than to that of CatL, suggesting that hBD3 can more potently inhibit the enzymatic activity of CatB than CatL. This is consistent with our previous observations that hBD3 (1 µM) inhibits the enzymatic activities of human recombinant CatB and CatL by approximately 60% and 10%, respectively [19]. It was reported that hBD3 can be a substrate for cysteine cathepsins such as CatB and CatL [35]. Therefore, the present findings extend our previous implication that hBD3 suppresses *Pg* LPS-induced oxidative and inflammatory responses in microglia through the suicide substrate-based inhibition of CatB and CatL. 

The methyl ester of CA-074Me is considered to be hydrolyzed by intracellular esterases releasing the active inhibitor, CA-074. However, the amount of CA-074 released into BV-2 microglia after treatment with CA-074Me remains unclear. Activity-based cysteine protease labeling using DCG-04 showed that CA-074Me is not a CatB-specific inhibitor in murine fibroblasts [36]. Therefore, CA-074Me may inhibit CatB and other cysteine cathepsins, especially CatL, at the concentrations used in previous studies (e.g., 10–30 µM). In contrast, ZRLR is a highly potent, irreversible, membrane-permeable, CatB-specific inhibitor. ZRLR is an azapeptide, which are peptide analogs in which the α-CH group of one amino acid resides in the peptide and is replaced by a nitrogen atom. The intrinsic structural prevalence of the *N*-terminal part of ZRLR is to remain bent, which allows a covalent attachment of ZRLR onto the active site Cys29 in CatB [37]. ZRLR exclusively blocks CatB in living primary antigen-presenting cells, in contrast to CA-074Me, by use of DCG-4, which detects active cysteine cathepsins in an activity- based reaction [32]. 

Currently, OMVs are considered to be potent vehicles for transmitting virulence factors into the host cells [38]. OMVs contain gingipains, which covalently bind to anionic LPS on their surface [31]. Furthermore, gingipains can activate pro-caspase-1 [39]. However, it is unlikely that the enzymatic activities of gingipains are involved in the IL-1β production, as pharmacological inhibition of both Kgp and Rgp did not block OMV-induced IL-1β production by BV-2 microglia. Our observations here showed that BV-2 microglia did not phagocytose OMVs prepared from KDP136, which is devoid of both Kgp and Rgp. It was reported that Rgp is necessary for the fimbriae formation [40,41]. Therefore, BV-2 microglia could not phagocytose OMVs prepared from KDP136 because of a lack of fimbriae, which is necessary for a lipid raft-mediated endocytic pathway [29,42]. Of particular note, OMVs prepared from KDP136 did not induce the IL-1β production by BV-2 microglia. 

OMVs secreted by Gram-negative bacteria function as a vehicle that delivers LPS into the cytosol and induces caspase-11-dependent inflammatory responses [43,44]. It has been suggested that a functional interaction between caspase-11 and the NLRP3 inflammasome, and perhaps involving additional partners, could promote noncanonical activation of the pro-IL-β processing [45]. It has been reported that phagocytic machinery can activate NF-κB in macrophages [46] and monocytes [47]. We previously reported that OMVs can deliver gingipains into the cytosol of hCMEC/D3 cells [27]. However, whether or not *Pg* LPS released into the cytosol is involved in the IL-1β production by BV-2 microglia after phagocytosis of OMVs remains unclear. The precise CatB-/CatL-independent mechanisms involved in OMV-induced IL-1β production by BV-2 microglia should be elucidated in future studies.

The limitation of tis study is that all experimental findings were obtained from in vitro studies by use of BV-2 microglia. Futher studies are also necessary to examine the effects of hBD3 (e.g., through intranasary injection) on AD-like pathological changes induced by chronic systemic exposure to *Pg* LPS and OMVs in middle-aged mice.

## 5. Conclusions

The abnormal expression and/or regulation of salivary antimicrobial peptides has been suggested to be associated with AD [48]. The present results suggest that hBD3, a salivary antimicrobial peptide, can improve the IL-1β-associated vicious inflammatory cycle induced by *Pg* LPS-stimulated microglia. Therefore, hBD3 may be a potential pharmacological intervention for the treatment of patients with sporadic AD, especially those with severe periodontitis.

## Figures and Tables

**Figure 1 cells-13-00283-f001:**
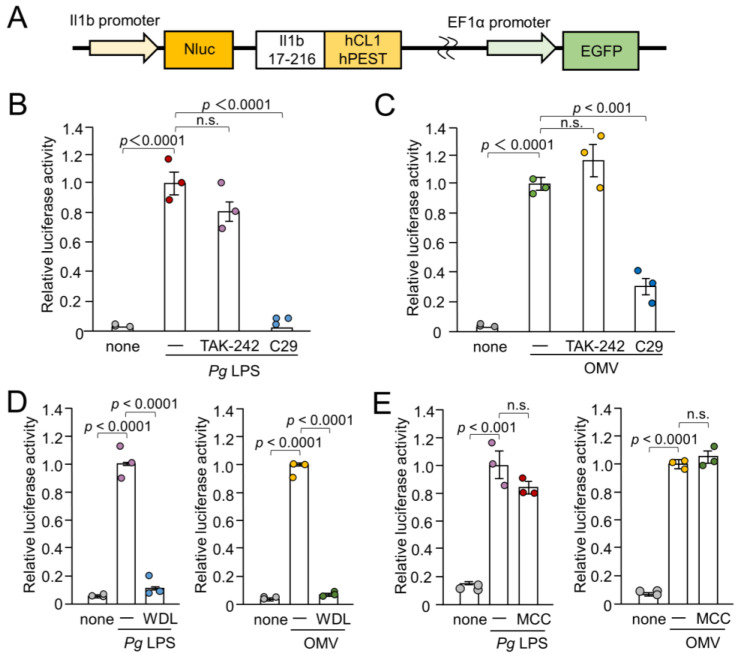
Schematic illustration of the IL-1β probe construct and effects of inhibitors for TLR4, TLR2, IKK, and NLRP3 inflammasome on the luciferase activity of the IL-1β probe in BV-2 microglia following stimulation with *Pg* LPS and OMVs for 1 h. (**A**) Nluc luciferase gene was fused with the sequence of *Il1b* cleaving site (*Il1b* 17-216). The *C*-terminus of the *Il1b* cleaving site was fused with two protein destabilization sequences (hCL1 and hPEST). The fusion gene was ligated downstream of the mouse IL-1β promoter. (**B**,**C**) The mean relative luciferase activity of the IL-1β probe induced by *Pg*LPS (**B**) and OMVs (**C**) in the absence or presence of the TLR4 inhibitor TAK-242 or the TLR2 inhibitor C29. (**D**) The mean relative luciferase activity of the IL-1β probe induced by *Pg* LPS and OMVs in the absence or presence of IKK inhibitor wedelolactone (WDL). (**E**). The mean relative luciferase activity of the IL-1β probe induced by *Pg* LPS and OMVs in the absence or presence of the NLRP3 inflammasome inhibitor MCC950 (MCC). The data relative to the values in *Pg* LPS or OMV-treated cells are presented as the mean ± SE of three independent experiments.

**Figure 2 cells-13-00283-f002:**
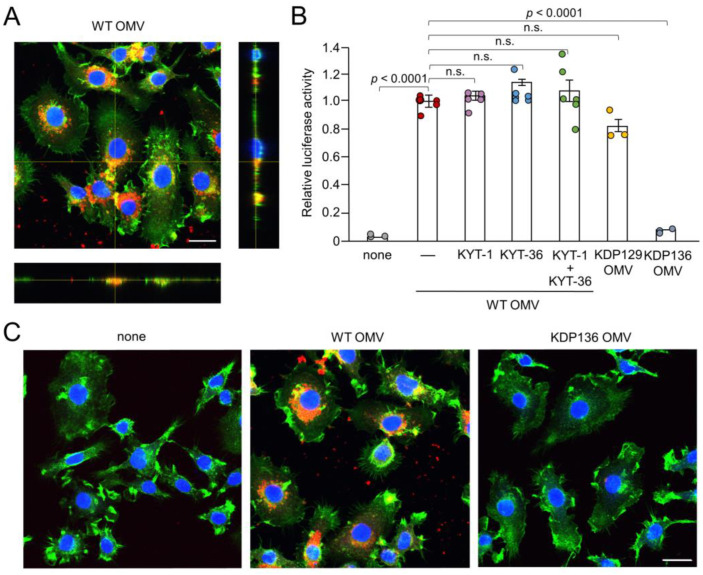
Phagocytosis of OMVs by BV-2 microglia and possible involvement of gingipains in OMV-induced IL-1β production. (**A**) CLSM images of BV-2 microglia after treatment with Cy5-labelled OMVs for 1 h prepared from wild-type strain (WT OMV). F-actin and nuclei were visualized with Acti-stain 488 phalloidin (green) and Hoechst 33342 (blue), respectively. Bottom and right rectangular panels represent z-stack images. Scale bar = 20 μm. (**B**) The mean relative luciferase activity of the IL-1β probe induced by OMVs for 1 h after pharmacological and genetic inhibition of gingipains. KYT-1: Rgp inhibitor; KYT-136: Kgp inhibitor; KDP129: OMVs prepared from Kgp mutant strain; KDP136: OMVs prepared from Rgp and Kgp mutant strains. The data relative to the values in WT OMV-treated cells are presented as the mean ± SE of three-six independent experiments. (**C**) CLSM images of BV-2 microglia after treatment with Cy5-labelled OMVs (red) prepared from wild-type (WT OMV) and gingipain-null mutant KDP136 strains for 1 h. F-actin and nuclei were visualized with Acti-stain 488 phalloidin (green) and Hoechst 33342 (blue), respectively. Scale bar = 20 μm.

**Figure 3 cells-13-00283-f003:**
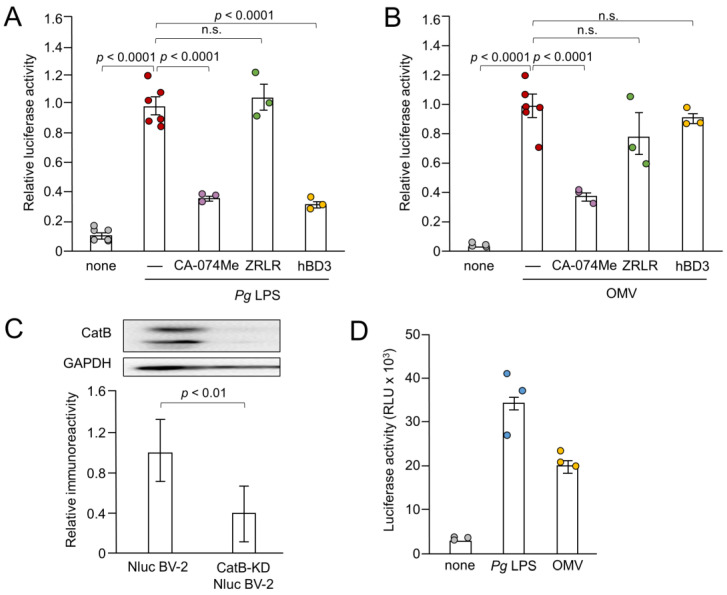
Effects of pharmacological and genetic inhibition of CatB on IL-1β production by BV-2 microglia following stimulation with *Pg* LPS and OMVs. (**A**,**B**) The mean relative luciferase activity of the IL-1β probe induced in BV-2 microglia following treatment with *Pg* LPS (**A**) and OMVs (**B**) after treatment with hBD3 (1 μM), CA-074Me (10 μM) or ZRLR (10 μM). The data are presented as the mean ± SE of 3-6 independent experiments. (**C**) The mean values of CatB intensity, which were detected by the immunoblot shown, were measured in Nluc reporter BV-2 microglia (Nluc BV-2) and Nluc reporter CatB-knockdown BV-2 microglia (CatB-KD Nluc BV-2) and normalized against the signal of GAPDH. The data are presented as the mean ± SE of three independent experiments, and the *p*-value was calculated using Student’s *t*-test. (**D**) The mean luciferase activity (RLU) of the IL-1β probe induced by *Pg* LPS or OMV in CatB-KD Nuc BV-2 microglia. The data are presented as the mean ± SE of 3 independent experiments.

**Figure 4 cells-13-00283-f004:**
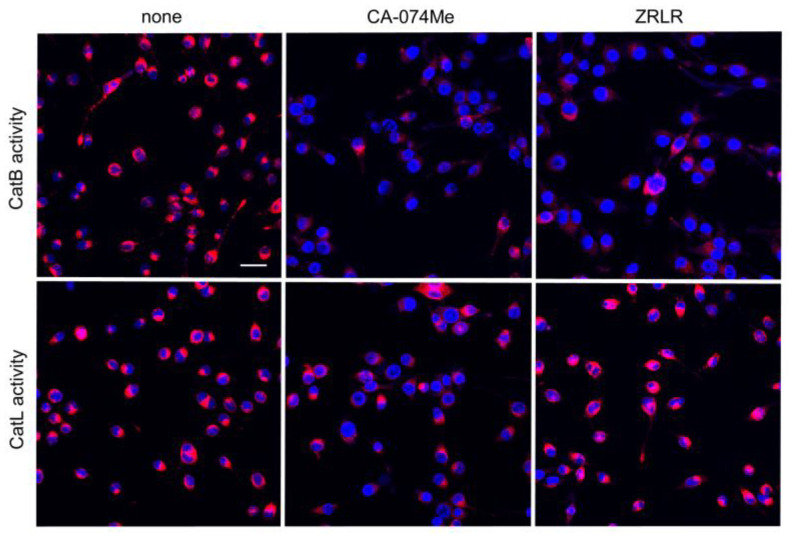
Enzymatic activities of CatB and CatL visualized using the cell-permeable, fluorescently labeled substrates, z-Arg-Arg-cresyl violet and z-Phe-Arg-cresyl violet, respectively, in the absence (none) and presence of CA-074Me (10 μM) or ZRLR (10 μM). Scale bar = 40 μm.

**Figure 5 cells-13-00283-f005:**
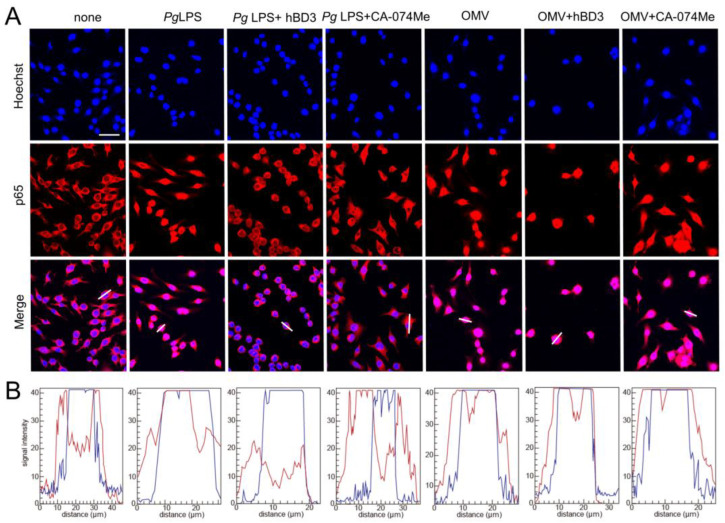
Effects of hBD3 and CA-074Me on nuclear NF-κB p65 translocation following stimulation with *Pg* LPS and OMVs. (**A**) Immunofluorescent CLSM images of BV-2 microglia after treatment with *Pg* LPS or OMVs in the absence or presence of hBD3 (1 μM) or CA-074Me (30 μM). Nuclear NF-κB p65 translocation was visualized by immunohistochemical staining (red). Nuclei were stained blue by Hoechst 33342 (blue). Scale bar = 40 μm. (**B**) The typical cells were analyzed by line plot profile to show the cytosol and nuclear NF-κB p65 translocation. The fluorescence intensity of NF-κB p65 and Hoechst 33342 in the cells traversed by white lines in (**A**) was indicated by red and blue lines, respectively.

**Figure 6 cells-13-00283-f006:**
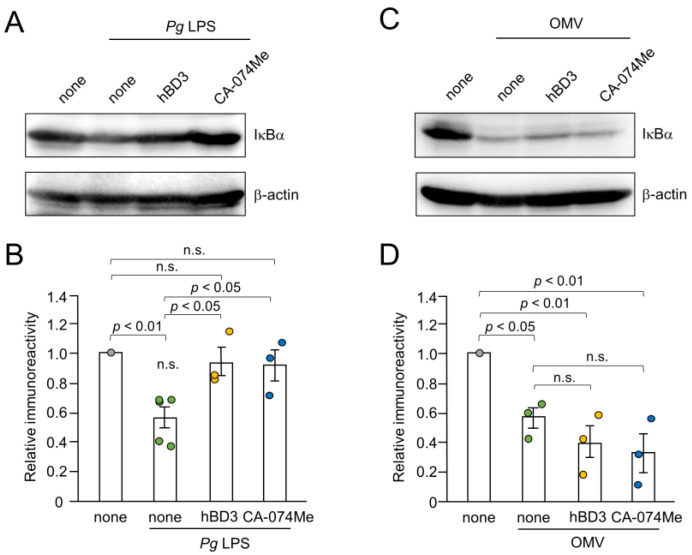
Effects of hBD3 and CA-074Me on the degradation of IκBα after stimulation with *Pg* LPS and OMVs. (**A**) The protein level of IκBα in BV-2 microglia after stimulation with *Pg* LPS (10 μg/mL) for 30 min in the presence or absence of hBD3 (1 μM) or CA-074Me (30 μM). (**B**) The mean values of the IκBα intensity shown in (**A**) were measured and normalized against the signal of β-actin. The data relative to the values in untreated cells are presented as the mean ± SE of three–five independent experiments. (**C**) The protein level of IκBα in BV-2 microglia after stimulation with OMVs (150 μg/mL) for 10 min in the presence or absence of hBD3 (1 μM) or CA-074Me (30 μM). (**D**) The mean values of the IκBα intensity shown in (**C**) were measured and normalized against the signal of β-actin. The data relative to the values in untreated cells are presented as the mean ± SE of three independent experiments.

**Figure 7 cells-13-00283-f007:**
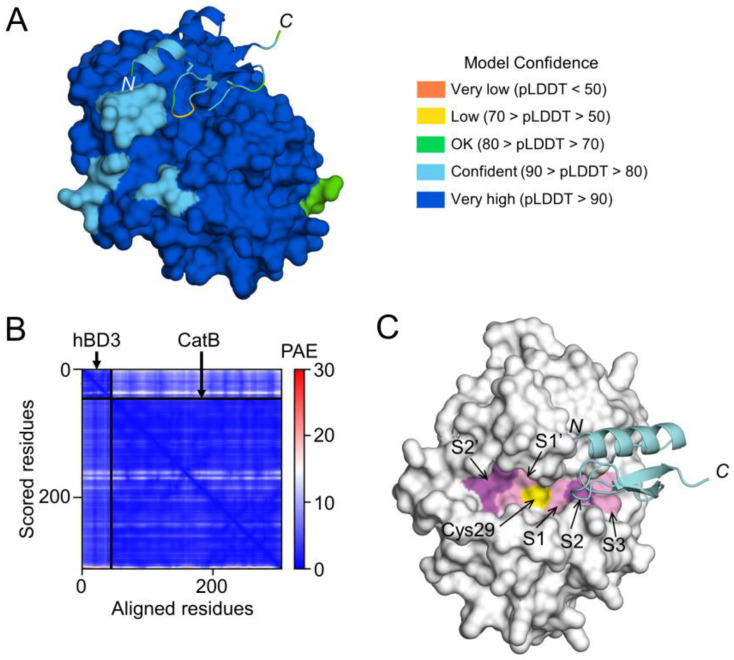
Prediction of hBD3 binding to CatB. (**A**) Structural model of hBD3-bound CatB generated using AlphaFold. The hBD3 model, presented as a ribbon, binds to the molecular surface of CatB. Amino acids are colored based on their pLDDT score. (**B**) PAE plots of the hBD3 and CatB complex model. (**C**) Binding of hBD3 to the active cleft formed on the molecular surface of CatB. The surface representation of CatB is shown in gray, except for the S1, S3, and S1’ sites, which are shown in light magenta; the S2 and S2’ sites, which are shown in magenta; and Cys29, which is shown in yellow. The hBD3 model, presented as a ribbon, is colored cyan. The figures were drawn using the PyMOL software program [33].

**Figure 8 cells-13-00283-f008:**
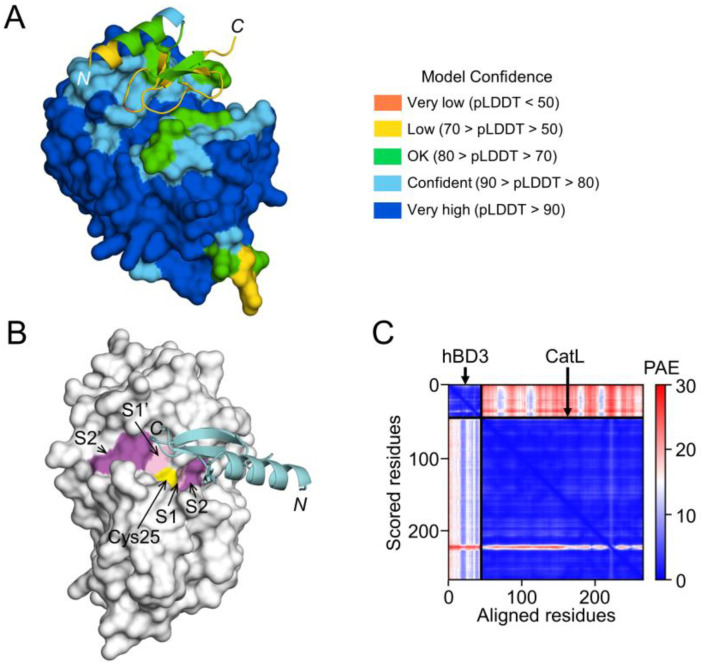
Prediction of hBD3 binding to CatL. (**A**) Structural model of hBD3-bound CatL generated using AlphaFold. The hBD3 model, presented as a ribbon, binds to the molecular surface of CatL. Amino acids are colored based on their pLDDT score. (**B**) Binding of hBD3 to the active cleft formed on the molecular surface of CatL. The surface representation of CatL is shown in gray, except for S1 and S1’sites, which are shown in light magenta; the S2 and S2’ sites, which are shown in magenta; and Cys25, which is shown in yellow. The hBD3 model, presented as a ribbon, is colored cyan. (**C**) PAE plots of the hBD3 and CatL complex model. The figures were drawn using the PyMOL software program [33].

**Table 1 cells-13-00283-t001:** Summary of effects of inhibitors on the IL-1β production, NF-κB p65 nuclear translocation and IκBα degradation by BV-2 microglia after stimulation with *Pg* LPS and OMVs.

**Name**	TAK-292	C29	WDL	MCC950	CA074Me	ZRLR	hBD3
**Target of Inhibitor**	TLR4	TLR2	IKK	NLRP3	CatB, CatL	CatB	CatB, CatL
***Pg* LPS**							
IL-1β production	−	+	+	−	+	−	+
NF-κB p65 traslocation	n.a.	n.a.	n.a.	n.a.	+	n.a.	+
IκBα degradation	n.a.	n.a.	n.a.	n.a.	+	n.a.	+
**OMVs**							
IL-1β production	−	+	+	−	+	−	−
NF-κB p65 traslocation	n.a.	n.a.	n.a.	n.a.	−	n.a.	−
IκBα degradation	n.a.	n.a.	n.a.	n.a.	−	n.a.	−

+: effective, −: not effective, n.a.: not analysed.

## Data Availability

The data that support the findings of this study are available from the corresponding author (H.N.), upon reasonable request.

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
