# Peer review of "Human β-Defensin 3 Inhibition of P. gingivalis LPS-Induced IL-1β Production by BV-2 Microglia through Suppression of Cathepsins B and L"

_cells, 2024, doi:10.3390/cells13030283_

Round 1

Reviewer 1 Report

Comments and Suggestions for Authors

The authors address the roles of CatB in Porphyromonas gingivalis lipopolysaccharide virulence factor-induced IL-1β production by microglia. Aim is to clarify effects of hBD3 and CatB inhibitors on the IL-1β production by microglia following stimulation with Pg LPS and OMVs.

In addition is could be suggested that one examines the role of oral antimicrobials on Porphyromonas gingivalis OMV content as I suspect the actions of stress will inevitably change the content (akin to antibiotics). Authors might want to include the impact of stress in the discussion.

The authors use stomach injection, does this relate to gum disease, please address the limitations of this approach why not inject the gums. The concentration in the liver is to be expected. Please address these limitations in the discussion.

The author further addresses the specific role of LPS and OMVs and AMPs on a proinflammatory cytokine, although one was investigated. Represents an additional step in overall understanding regarding contributing factors to Alzheimers.

Include the size distribution of the OMVs as before.

 Explanation as to why the concentrations was set 1 micromolar for the AMP compared to 10 micromolar for the controls.

 Overall, the main questions posited at the end of the introduction were addressed practically through experimentation and theoretically modelling.

References are little outdated peaking in 2015 and 2020.

Legend figure 6 needs to be rewritten tagged with plagiarism 20%.

Expansion of the discussion is needed regarding limitations of experimental approach the role of stress on OMV content, and the role of LPS in Alzheimers should be identified, Alternative Routes of administration need discussion.

Author Response

  1. The authors address the roles of CatB in Porphyromonas gingivalis lipopolysaccharide virulence factor-induced IL-1β production by microglia. Aim is to clarify effects of hBD3 and CatB inhibitors on the IL-1β production by microglia following stimulation with Pg LPS and OMVs. In addition is could be suggested that one examines the role of oral antimicrobials on Porphyromonas gingivalis OMV content as I suspect the actions of stress will inevitably change the content (akin to antibiotics). Authors might want to include the impact of stress in the discussion.

[Author’s reply]

       Thank you for this suggestion. However, we must note that the impact of stress is not a central theme in this study. We’d like to examine the impact of stress on the P. gingivalis OMV content in future studies.

  1. The authors use stomach injection, does this relate to gum disease, please address the limitations of this approach why not inject the gums. The concentration in the liver is to be expected. Please address these limitations in the discussion.

[Author’s Reply]

       In the present study, we conducted in vitro experiments using BV-2 microglia. So, we did not use stomach injection.

  1. The author further addresses the specific role of LPS and OMVs and AMPs on a proinflammatory cytokine, although one was investigated. Represents an additional step in overall understanding regarding contributing factors to Alzheimers. Include the size distribution of the OMVs as before.

[Author’s Reply]

       According to the reviewer’s comment, we have represented an additional step in overall understanding regarding contributing factors to Alzheimer’s disease in the Discussion section as follows:

       The limitation of this study is that all experimental findings were obtained from in vitro studies by use of BV-2 microglia. Further studies are also necessary to examine the effects of hBD3 (e.g. through intranasary injection) on AD-like pathological changes induced by chronic systemic exposure to Pg LPS and OMVs in middle-aged mice.

       In this study, we did not measure the diameter of Pg OMVs. In general, the diameter of Pg OMVs are 50-250 nm (Xie H, Future Micobial, 10, 1517-1527, 2015). I our experiments, we probably used 50-220 nm of Pg OMVs, because we passed Pg supernatant through the membrane filter with pore size of 220 nm before preparation of OMVs from Pg supernatant by ultracentrifugation and Total Exosome Isolation Reagent.

  1. Explanation as to why the concentrations was set 1 micromolar for the AMP compared to 10 micromolar for the controls.

[Author’s Reply]

       In our previous study (Inoue et al., In J Mol Sci. 2022, 23, 15099), human β-defensin 3 did not exhibit a significant toxic effect on MG6 cells up to 1 μM, but 10 μM human β-defensin 3 significantly reduced the cell viability. Therefore, we have decided use 1 μM human β-defensin 3 in this study. On the other hand, CA-074Me showed significant inhibitory effect with the concentration of 10-30 μM without any  cytotoxicity, but not 1 μM,

  1. References are little outdated peaking in 2015 and 2020.

[Author’s Reply]

       We appreciate this comments. According to the reviewer’s comment, we have replaced Kadowaki et al. (J Biol Chem, 1998) and Muroi et al. (J Biol Chem., 1994) with Kadowaki (Methods Mol Biol. 2021) and Lee et al. (BMB Rep., 2016), respectively.

[39] Kadowaki T. Enzymatic characteristics and activities of gingipains from Porphyromonas gingivalis. Methods Mol Biol. 2021, 2210, 97-112.

[45] Lee KJ, Kim YK, Krupa M, Nguyen AN, Do BH, Chung B, Vu TT, Kim SC, Choe H. Crotamine stimulates phagocytic activity by inducing nitric oxide and TNF-α  via p38 and NF-κB signaling in RAW264.7 macrophages. BMB Rep. 2016, 49, 185-190. 

  1. Legend figure 6 needs to be rewritten tagged with plagiarism 20%.

[Author’s Reply]

       According to the reviewer’s comment, We have reconstructed sentences of the Figure 6 legend as follows:

Figure 6. Effects of hBD3 and CA-074Me on the degradation of IκBα after stimulation with Pg LPS and OMVs. (A) The protein level of IκBα in BV-2 microglia after stimulation with Pg LPS (10 μg/mL) for 30 min in the presence or absence of hBD3 (1 μM) or CA-074Me (30 μM). (B) The mean values of the IκBα intensity shown in (A), were measured and normalized against the signal of β-actin. The data relative to the values in untreated cells are presented as the mean ± SE of three-five independent experiments. (C) The protein level of IκBα in BV-2 microglia after stimulation with OMVs (150 μg/mL) for 30 min in the presence or absence of hBD3 (1 μM) or CA-074Me (30 μM). (D) The mean values of the IκBα intensity shown in (C), were measured and normalized against the signal of β-actin. The data relative to the values in untreated cells are presented as the mean ± SE of three independent experiments.

  1. Expansion of the discussion is needed regarding limitations of experimental approach the role of stress on OMV content, and the role of LPS in Alzheimers should be identified, Alternative Routes of administration need discussion.

[Author’s Reply]

       Thank you for these comments. However, as mentioned before, the impact of stress on Porphyromonas gingivalis OMV contents is out of the scope of this study. For a possible pathological of LPS in Alzheimer’s disease, we have described in the Introduction as follows:

       Furthermore, we first reported that chronic systemic exposure to Pg LPS induces AD-like pathologies, including microglia-mediated neuroinflammation and cognitive decline in middle-aged mice, but not in CatB-deficient mice [20].

Reviewer 2 Report

Comments and Suggestions for Authors

The authors describes the results of their experiments with oral bacteria LPS and outer membrane vesicles (OMV) on BV-1 microglia. They use a number of specific inhibitors to evaluate the possible pathways of these agents in IL-1β production of microglia. As a result they find differences in the activation pathway between LPS and OMV effect. In addition, they describe the inhibitory effect of human defensing via cathepsins. The work is carried out by well planned but not always clearly described experiments with modern techniques.

The title is too long and not easy to understand at the first sight. It needs some background information why an oral pathogen was chosen for these studies. (Just as minor remark: could it be possible that patients with Alzheimer’s disease have worse oral hygiene and that is the reason of the appearance of that bacterium in the brain?) A figure of the reporter plasmid used in the BV-2 cells can help to understand how it works (EGFP positive cell selection). It would be helpful to describe in a sentence the mutant bacterial strains. What are the functions of the OMVs in general? Do they have effect without LPS? Do they carry proteases? The authors used two types of CatB inhibitors in the experiment with different results. What can be the cause of it?

In 3.4 it is claimed that the inhibition of CatB reduced the LPS generated IL-1β secretion. Why did not the shRNA method for CatB expression reduction give the same result? In reference #35 Taggart et al. described the interaction of beta-defensine and cathepsins, but with opposite action. What can be the cause of this difference.

It would be helpful to summarize (in a table?) all the inhibitors used in the experiments, their results and the consequences of them.  

Author Response

  1. The title is too long and not easy to understand at the first sight.

[Author’s Reply]

       According to the reviewer’s comment, we have revised the title to “Human β-defensin 3 Inhibition of P. gingivalis LPS-induced IL-1β Production by BV-2 Microglia Through Suppression of Cathepsins B and L”.

  1. It needs some background information why an oral pathogen was chosen for these studies. (Just as minor remark: could it be possible that patients with Alzheimer’s disease have worse oral hygiene and that is the reason of the appearance of that bacterium in the brain?)

[Author’s Reply]

       In the Introduction section of our previous paper (Inoue et al. In J Mol Sci. 2022, 23, 15099), we described a detailed information about periodontitis as a risk factor of Alzheimer’s disease. In order to avoid duplication, we have kept a minimum description of the relationship between Alzheimer’s disease and virulence factors of P. gingivalisin the Introduction section of the present study as follows:

       The emerging role of microbes and innate immune pathways in AD pathology suggests that antimicrobial peptides may be effective as early therapeutic intervention in future clinical trials [16, 17].

       Furthermore, we first reported that chronic systemic exposure to Pg LPS induces AD-like pathologies, including microglia-mediated neuroinflammation and cognitive decline in middle-aged mice, but not in CatB-deficient mice [20].

  1. A figure of the reporter plasmid used in the BV-2 cells can help to understand how it works (EGFP positive cell selection). It would be helpful to describe in a sentence the mutant bacterial strains.

[Author’s Reply]

       According to the reviewer’s comment, we have added a schematic illustration of the IL-1β  probe construct in the revised Figure 1A.

       We have also added a more detailed description of mutant P. gingivalisin the Material and Methods section as follows:

      There are three types of gingipains: two arginine-specific proteases (RgpA and RgpB) and a lysine-specific protease (Kgp).Pg ATCC33277 (wild-type; WT), Kgp-deficient mutant strain (KDP129), and RgpA, RgpB and Kgp-deficient mutant strain (KDP136) were maintained as described previously [27].

  1. What are the functions of the OMVs in general? Do they have effect without LPS? Do they carry proteases?

[Author’s Reply]

       We have described a pathological function of OMVs and components of OMVs in the Discussion section as follows:

       Currently, OMVs are considered to be potent vehicles for transmitting virulence factors into the host cells [37]. OMVs contain gingipains, which covalently bind to anionic LPS on their surface [31].

  1. The authors used two types of CatB inhibitors in the experiment with different results. What can be the cause of it?

[Author’s Reply]

       ZRLR is a membrane-permeable specific inhibitor of cathepsin B. On the other hand, CA-074Me is known to inhibit cathepsin B as well as cathepsin L (Reich et al. 2009).

[32] Reich M, Wieczerzak E, Jankowska E, Palesch D, Boehm BM, Burster T. Specific cathepsin B inhibitor is cell-permeable and activates presentation of TTC in primary human dendritic cells. Immun Lett. 2009, 123, 155-159.

       We have described the differential inhibitory properties of ZRLRZ and CA-074Me in the Result section as follows:

       We examined the possible inhibitory effects of CA-074Me and ZRLR on the enzymatic activities of CatB and CatL in BV-2 microglia using cell-permeable, fluorescently labeled substrates, z-Arg-Arg-cresyl violet and z-Phe-Arg-cresyl violet, respectively. The fluorescent cresyl violet group was designed to be dequenched upon cleavage of dipeptides by CatB or CatL. CA-074Me (10 µM) markedly reduced the enzymatic activity of both CatB and CatL in BV-2 microglia. In contrast, ZRLR (10 µM) markedly reduced the fluorescent signal of CatB, but not that of CatL (Figure 4).

  1. In 3.4 it is claimed that the inhibition of CatB reduced the LPS generated IL-1β secretion. Why did not the shRNA method for CatB expression reduction give the same result?

[Author’s Reply

       ZRLR, a membrane-permeable specific inhibitor, showed the same results with the knockdown of cathepsin B by use of the shRNA method. They did not inhibit the LPS-induced IL-1β production. In contrast, both CA-074Me and β-defensin 3 significantly inhibited the LPS-induced IL-1β production, because they could inhibit the enzymatic activity of cathepsin B as well as cathepsin L.

  1. In reference #35 Taggart et al. described the interaction of beta-defensine and cathepsins, but with opposite action. What can be the cause of this difference.

[Author’s Reply]

       We appreciate the reviewer for the pertinent comment. It is well known that substrates with high concentrations can inhibit the enzyme activity. Taggart et al. used human β-defensin 3 with the concentration of 19 nM, while we used 1 µM human β-defensin 3. The descrepanncy may mainly due to the differential concentration used. Furthermore, the present AlphaFold models further indicate that human β-defensin 3 can strongly bind to the active site of cathepsin B. Human β-defensin 3 also binds to the active site of cathepsin L. Therefore, we consider that human b-defensin 3 inhibits the enzymatic activities of both cathepsins B and L through a suicide substrate-based inhibition.

  1. It would be helpful to summarize (in a table?) all the inhibitors used in the experiments, their results and the consequences of them.

[Author’s Reply]

       According to the reviewer’s comment, we have summarized all the inhibitors used in the experiments, their results and the consequences of them in Table 1.

Round 2

Reviewer 2 Report

Comments and Suggestions for Authors

I accept the answers and modifications of the paper.